# Circulating miRNAs and Preeclampsia: From Implantation to Epigenetics [note 1]

**DOI:** 10.3390/ijms25031418

**Published:** 2024-01-24

**Authors:** Stefano Raffaele Giannubilo, Monia Cecati, Daniela Marzioni, Andrea Ciavattini

**Affiliations:** 1Department of Clinical Sciences, Università Politecnica delle Marche, 60020 Ancona, Italy; s.giannubilo@univpm.it (S.R.G.); a.ciavattini@staff.univpm.it (A.C.); 2Department of Experimental and Clinical Medicine, Università Politecnica delle Marche, 60126 Ancona, Italy; d.marzioni@staff.univpm.it

**Keywords:** miRNA, preeclampsia, trophoblast, placenta, epigenetic

## Abstract

In this review, we comprehensively present the literature on circulating microRNAs (miRNAs) associated with preeclampsia, a pregnancy-specific disease considered the primary reason for maternal and fetal mortality and morbidity. miRNAs are single-stranded non-coding RNAs, 20–24 nt long, which control mRNA expression. Changes in miRNA expression can induce a variation in the relative mRNA level and influence cellular homeostasis, and the strong presence of miRNAs in all body fluids has made them useful biomarkers of several diseases. Preeclampsia is a multifactorial disease, but the etiopathogenesis remains unclear. The functions of trophoblasts, including differentiation, proliferation, migration, invasion and apoptosis, are essential for a successful pregnancy. During the early stages of placental development, trophoblasts are strictly regulated by several molecular pathways; however, an imbalance in these molecular pathways can lead to severe placental lesions and pregnancy complications. We then discuss the role of miRNAs in trophoblast invasion and in the pathogenesis, diagnosis and prediction of preeclampsia. We also discuss the potential role of miRNAs from an epigenetic perspective with possible future therapeutic implications.

## 1. Introduction

The placenta is a transitory organ, necessary for development in utero in humans and mammals [1,2,3]. The important role of this organ is highlighted when normal placental development is compromised, leading to important pregnancy complications, such as preeclampsia (PE) [4,5], fetal growth restriction (FGR) [6,7,8], gestational diabetes mellitus (GDM) [9], preterm birth (PTB) [9,10] and gestational trophoblastic disease (GTD) [11,12].

Preeclampsia is a hypertensive disorder associated with pregnancy that affects 5–7% of pregnancies and is responsible for over 70,000 maternal deaths each year [13]. Parenteral or oral drug therapies are available and administered to preeclamptic women to stabilize the mother and fetus and reduce the risk of failure of several maternal organs, such as the liver, kidney and brain [14]. Nevertheless, to date, delivery at an optimal time is the only reliable treatment for preeclampsia [15]. 

Reduced arterial compliance and peripheral vasoconstriction characterize the preeclamptic condition from the early stages [16]; however, the exact pathophysiology of preeclampsia is still unknown. Probably, the initial defect in placentation and vascularization in the placenta bed due to poor trophoblast invasion of the spiral arteries [17,18,19] leads to an inappropriate activation of the immune system [20], as it occurs in recurrent miscarriage [21]. It is known that oxidative stress and inflammation are involved in endothelial dysfunction [22,23,24,25,26,27], a key process underlying several diseases including preeclampsia [28,29,30,31,32]. Hypoxia due to inadequate spiral artery remodeling compromises the placental endothelial function [33], which results in an oxidative stress condition [34,35], made worse by an increased production of nitric oxide [36,37]. Continued oxidative stress leads to chronic inflammation, which induces a premature senescence of preeclamptic placental tissue, as suggested by the short length of telomeric DNA [38]. 

The inability of preeclamptic placental tissue to regain a condition of equilibrium leads to an inevitable dysregulation of several metabolic pathways [39,40]. The understanding of this pathology is still elusive, although different studies tried to identify candidate genes involved in preeclampsia onset, measuring their expression in placental tissue [41,42]. 

Since preeclampsia can cause maternal and neonatal morbidity and mortality [43], several studies have been focused on finding efficient markers able to predict preeclampsia onset to improve the treatment of these pregnancies [5,44,45,46,47,48,49]. 

The presence of circulating RNAs released from many body tissues, the placenta included, offers a potential tool to indirectly observe any pathologies in real time, starting from the onset and during development. In this way, it possible to associate the physiological changes in the circulating RNA variation level in preeclamptic subjects with respect to healthy ones [50]. Circulating RNAs contain fragments that are transcribed but not translated called non-coding RNAs (ncRNAs). Bioinformatic analysis estimates that a quote of the human genome equal to 75–80% is transcribed into RNA molecules. Nevertheless, just 2% of transcribed RNA molecules are converted into proteins [51]. Usually, ncRNAs are divided into four different classes: PIWI-interacting RNAs (piRNAs), circular RNAs (circRNAs), long ncRNAs (lncRNAs) with a length over 200 nucleotides and small ncRNAs (known also as miRNAs) with a length under 200 nucleotides. 

The name piRNAs was derived from their ability to bind to PIWI family proteins. They are between 24 and 30 nucleotides in length, and they are known to participate in the regulation of cell proliferation, apoptosis and the cell cycle [52]. LncRNAs and circRNAs are more than 200 nucleotides in length but have different shapes. As suggested by their name, lncRNAs are linear unlike circRNAs, which are ringlike. Both lncRNAs and circRNAs are transcribed from regions located at exons, introns, intergenic regions or 5′/3′-untranslational regions and interact with DNA, RNA and proteins by folding themselves into complicated second structures [53]. LncRNAs and circRNAs can modulate gene expression through several strategies. They can prevent mRNA degradation, avoiding the miRNA, by bind to the targeted mRNA. They can regulate gene expression by modulating transcription factors to tie to promoters [54]. In addition, they can exert a regulative action on several signaling pathways, acting as a scaffold to regulate the interactions between proteins [55]. miRNAs are small ncRNAs, which control mRNA expression. Changes in miRNA expression can induce a variation in the relative mRNA level and influence cellular homeostasis [56,57]. Many miRNAs remain in the cell where they were assembled, whereas others can be released outside (known as circulating miRNAs) under normal and pathological conditions [58]. The strong presence of circulating miRNAs in all body fluids makes them useful biomarkers of several diseases, cancer included [44,59,60]. Circulating miRNAs are present in gastric juices, saliva and urine [61,62,63], but plasma and serum represent the source most frequently used to collect them in healthy and non-healthy subjects [64].

This review summarized the miRNAs recovered from the peripheral maternal blood circulation that resulted in a dysregulated preeclamptic pregnancy from its onset through to its development. The authors also focused on the epigenetic regulations of miRNA expression as a possible tool to reduce the risk of cardiovascular diseases in women who experienced preeclamptic pregnancy.

## 2. Materials and Methods

During preparation of this scoping review (September to December 2023), we searched PubMed using terms containing “miRNA”, “pregnancy”, “preeclampsia”, “hypertension”, and “Placenta”. Only studies on circulating miRNAs were included. We also identified relevant studies via ‘snowballing’/citation chasing that were relevant for background information. All studies were assessed based on their quality of reporting, bias in participant selection, result presentation or author conflicts of interest.

## 3. miRNA Biogenesis, Mechanisms of Export and Annotation Criteria

miRNAs are single-stranded non-coding RNAs, 20–24 nt long, which originate from the primary miRNA transcript (pri-miRNA) [65]. In the nucleus, an RNase III endonuclease (named Drosha) combined to DGCR8 (DiGeorge critical region 8) protein (named Pasha) begins the maturation of the pri-miRNA, liberating a ~60–70-nt stem loop intermediate (pre-miRNA). The complex Exportin-5–Ran-GTP moves the pre-miRNA precursor hairpin from the nucleus to the cytoplasm. The RNase Dicer associated with the transactivation response element RNA-binding protein (TRBP) digests the pre-miRNA to find its mature length. Finally, the passenger strand of the mature miRNA is digested, while the functional strand is loaded with Argonaute (Ago2) proteins into the RNA-induced silencing complex (RISC). RISC silences the mRNA target through mRNA deadenylation, translational repression or cleavage driven by the functional strand of the mature miRNA, which recognizes 3′-untranslated region (3′-UTR) [66,67]. Although the 3′UTR sequence is the most prevalent site for the recognition of mRNA target, miRNAs can also interact with 5′-untranslated region (5′-UTR) [68] (Figure 1).

In recent years, circulating miRNAs have gained the attention of activity research because they function in the same manner as classical signaling, made of growth factors, hormones and cytokines [69]. Circulating miRNAs are surprisingly stable and able to carry out their suppressive function against mRNA target into the recipient cells. Mitchell et al. [70] and Chen et al. [71] demonstrated that miRNAs isolated from human serum and plasma were resistant to activity of RNase. Contrarily, exogenous miRNAs synthetized to be added to plasma and serum resulted in rapid degradation, suggesting that the integrity of endogenous miRNA in human biological fluids depends on something different than their structure or short length [70]. 

Experimental data demonstrated that miRNAs can be released outside the cell under four different forms: (a) miRNAs combined to Argonaute2 protein (Ago2) [72,73], (b) miRNAs tied to RNA-binding protein nucleophosmin (NPM1) [74], (c) miRNAs tied to high-density lipoprotein (HDL) [75], and (d) miRNAs closed within extracellular vesicles (EVs) such as exosomes and microvesicles. Notably, the presence of one form of miRNA does not exclude the presence of another extracellular form of miRNAs, as suggested by results in all cell lines or human biological samples tested [76]. 

The circulating miRNAs detected in plasma, and not included in EV, are mostly bound to proteins. The Ago2 protein not only takes part in miRNA maturation via RISC complex but also exercises a shielding effect on extracellular miRNAs against RNases. Particularly, Arroyo et al. suggested that the entire Ago2–miRNA complex may be able to modulate gene expression in recipient cells [73]. 

Like Ago2, NPM1 also has a protective role for circulating miRNAs. Wang et al. hypothesized that NPM1 is involved not only in the packaging and release of miRNAs outside the cell but that it remains bound to them in the peripheral circulation [74]. Nevertheless, the biological function of circulating miRNAs associated with both Ago2 and NPM1 proteins in pregnancy and pregnancy-related diseases is still unknown.

HDL is one of the five different types of lipoproteins, including chylomicrons, very-low-density lipoprotein (VLDL), intermediate-density lipoprotein (IDL) and low-density lipoprotein (LDL). HDLs have a micellular constitution and organization, like most polar lipids, and a mass composed of 50–60% proteins, mostly represented by apolipoproteins A-I (APOA1) [74]. It is widely known that HDLs also vehicle miRNAs, not only lipids and proteins [75]. The load of miRNAs in HDFs is different in composition and concentration between normal and pathological subjects and equally attributable to a specific disease [77,78,79]. It has been demonstrated that the miRNAs-HDL complex is internalized only if the recipient cell expresses the scavenger receptor BI (SR-BI) [75]. Unfortunately, currently, no evidence is available in the literature about the biological importance of this mechanism in pregnancy and especially in the onset of pathological pregnancies. 

EVs are small lipid particles released from most human cell types, both malignant and healthy [80]. Interestingly, EVs have also been suggested as a promising class of therapeutic agents [81]. EVs mediate intercellular communication by moving heterogeneous molecules (i.e., DNA, RNAs, miRNAs, proteins and lipids). The recognition of miRNAs sorted for cargo into EVs relies on heterogeneous nuclear ribonucleoprotein A2B1 (hnRNPA2B1) or synaptotagmin-binding cytoplasmic RNA interaction protein (SYNCRIP). A paper published by Villarroya-Beltri et al. demonstrated that the selective choice of miRNAs to be released outside by EVs depends on the recognition of special sequences named EXOmotifs (e.g., GGAG or AGG) in miRNAs by hnRNPA2B1 or SYNCRIP [82]. The membranous protein caveolin-1 has been identified as necessary to drive the miRNAs-hnRNPA2B1/ SYNCRIP complex into EVs [83]. Once formed, EVs and their cargo can be liberated, either directly from the plasma membrane or grouped into multivesicular bodies and then released [81]. 

A passive release of miRNAs can occur in a state of chronic inflammation, apoptosis or necrosis, leading to cell lysis. In this case, the miRNAs are released into biological fluids associated with Ago proteins [72]. A quantitative analysis of the passive release of miRNAs has not yet been carried out.

miRNAs are identified by a code made of the “miR” prefix and a unique identifying number. miRNAs belonging to certain species are indicated with an additional three-letter prefix; for example, hsa-miR-124 is a miRNA belonging to the human species (the prefix hsa means Homo Sapiens). The identifying numbers are given sequentially, meaning that miR-124 was discovered and named before miR-456. miRNAs with nearly identical sequences, except for one or two nucleotides, can be given with an additional lowercase letter [84]. Let-7 is a fundamental miRNA tumor suppressor. In humans, the let-7 family includes 12 different members (let-7a-1, 7a-2, 7a-3, 7b, 7c, 7d, 7e, 7f-1, 7f-2, 7g, 7i and miR-98) [85]. The chromosome 19 miRNA cluster (C19MC) includes approximately 8% of all known human miRNA genes. C19MC is only expressed in the placenta and in undifferentiated cells [86].

## 4. miRNAs, Implantation and Preeclampsia

The possibility that miRNAs may play a crucial role in the pathogenesis of preeclampsia stems from the consideration that these molecules actively participate in the implantation processes of pregnancy and, on the other hand, that preeclampsia finds its pathophysiological basis in the very early stages of pregnancy. Nucleic acids have been detected in uterine fluid and, more specifically, in EVs containing miRNAs, suggesting a role of miRNAs in embryo–endometrial communication [87]. Using biopsy material and modern transcriptomics, it has also been shown that miRNAs are dysregulated in the endometrium of women with recurrent implantation failure [88]. Trophoblast migration and invasion, and cellular adaptations in the physiological changes underlying gestation, involve EVs as key modulators. Endometrial luminal epithelial cells [89] and proper communication between these cells determine the success or failure of pregnancy [90]. Exosome nanovesicles can transfer information (e.g., hsa-miR-30b, hsa-miR-200c, hsamiR-17 and hsa-miR-106a, miRNAs involved in endometrial receptivity and implantation) from the endometrium to the blastocyst, thereby promoting implantation [91,92]. In addition, extracellular vesicles secreted by the endometrium are internalized by the embryo to enhance adhesion and invasion [93], mediating embryo–endometrial communication [94]. During the early invasion phase (8 to 10 days after conception), the cytotrophoblast differentiates into an extravillous interstitial trophoblast and invades the decidua. At this point, feto–maternal communication occurs between the extravillous trophoblast and the decidualized endometrial stroma [95]. One of the most probable hypotheses to describe the etiology of preeclampsia is based on a failure of extravillous trophoblasts to invade the uterine spiral arteries in the placental bed. Insufficient placental vascular remodeling induces placental hypoperfusion, which is critical for the pathogenesis of preeclampsia. Whitley et al. showed that first-trimester extravillous trophoblasts from pregnancies with high uterine artery resistance were inherently more sensitive to apoptotic stimuli, which may be associated with reduced remodeling of the maternal spiral arteries [96]. Placental insufficiency has also been associated with abnormal levels of extracellular fetal DNA, mRNA transcripts and circulating C19MC microRNAs (miR-516b-5p, miR-517-5p, miR-520a-5p, miR-525-5p and miR-526a) [97]. Exosomes secreted by cytotrophoblasts, which express placenta-specific miRNAs, including syncytin-2, have been implicated in embryo implantation through the promotion of Treg differentiation and suppression of the nuclear factor B signaling pathway and, thus, the immune and inflammatory response [98]. The specific loading of miRNAs in maternal plasma exosomes obtained in the first trimester of pregnancy in women who developed preeclampsia [99] suggests a potential role of miRNAs in the pathogenesis of preeclampsia as early as the first trimester. For example, miR-1269 mediates the downregulation of the tumor suppressor gene forkhead box O1 (FOXO1) [100]. Because FOXO1 is a protein involved in the stromal decidualization of the endometrium and the implantation process [101], alterations in its expression can have deleterious consequences in pregnancy, such as preeclampsia. The expression level of miR-33b-3p in patients with preeclampsia is significantly different from that of healthy pregnant women. It has been reported that transforming growth factor beta-1 (TGFB1), a target of miR-33b-3p, may play a role in the pathogenesis of preeclampsia by preventing the differentiation of trophoblasts toward an invasive phenotype [102]. In other experimental studies, it was seen that the migratory and invasive abilities of trophoblastic cells were significantly inhibited by miR-486-5p. Rho GTPase-activating protein 5 (ARHGAP5) is the most abundant GTPase-activating protein (GAP) of the small Rho GTPase family. This protein plays a role in the regulation of actin cytoskeleton-based mechanisms, thereby influencing cell migration and invasion [103]. The possible pathogenesis of preeclampsia could depend on increased miR-486-5p, which modulates the expression of ARHGAP5 in trophoblast cells [104]. The X-linked inhibitor of apoptosis protein (XIAP) is an inhibitory protein of apoptosis (IAP) [105]; the expression of XIAP in trophoblasts is decreased in preeclampsia, which may be associated with increased apoptosis in the placenta [106]. Serum miR-515-5p was significantly increased in patients with preeclampsia compared with normal pregnant women. Functionally, the overexpression of miR-515-5p suppressed the proliferation and invasion of trophoblastic cells, as observed in HTR-8/SVneo trophoblastic cells. Using luciferase reporter assays, XIAP was identified as a direct target gene of miR-515-5p, and the overexpression of miR-515-5p was found to decrease the level of XIAP in HTR-8/SVneo cells [107]. Of particular interest is the relationship between miRNAs and oxygenation. Early in gestation (<12 weeks), the placenta develops in a low-oxygen environment (<20 mmHg), which is necessary for successful placentation [108]. The insufficient syncytization of cytotrophoblastic villus cells results in suboptimal placental perfusion and, thus, chronic hypoxia, which is a hallmark of preeclampsia. Placental renin–angiotensin system (RAS) activity depends, in part, on the prevailing oxygen tension, and the latter controls the levels of placental miRNAs that regulate placental RAS expression. The suppression of miRNAs targeting placental RAS early in gestation is partly responsible for the increase in RAS expression during this period to promote placental development [109]. In other studies in this area, it has been reported that miR455-3P acts as a rheostat that restrains the hypoxia response that might otherwise prevent the differentiation of the cytotrophoblast into syncytiotrophoblast: reduced miR455 expression is causally linked to the development of preeclampsia [110].

## 5. miRNAs and Clinical Preeclampsia

This section reports results from different studies about the possibility of using miRNAs as a tool for the prediction of preeclampsia onset and severity. Significantly dysregulated miRNAs are reported in Table 1.

### 5.1. miRNAs and Diagnosed Preeclampsia

One of the first demonstrations of the possibility of using circulating miRNAs as a marker for preeclamptic pregnancy was by Gunel et al. They measured the expression level of miR-210 and miR-152 from the maternal plasma of both healthy and preeclamptic pregnant women. The results demonstrated an upregulation for miR-210 and a downregulation for miR-152 in women with preeclamptic pregnancy [133]. In the study of Campos et al., a lower expression of circulating miR-196b in maternal plasma was correlated to a preeclamptic condition in pregnant women [138]. Both miR-195-5p [137] and miR-885-5p [150] overexpression was correlated to a preeclamptic condition in women. Investigating the plasma of women with diagnosed pregnancy complicated by preeclampsia, Sheng et al. observed an upregulation for miR-206 [140]. The study realized by Akgor et al. analyzed the circulating level of miRNAs in women with diagnosed pregnancy complicated by preeclampsia. The authors observed a panel of miRNAs, many of which are already known as potential biomarkers for the non-invasive diagnosis of preeclampsia. However, they observed, for the first time, an upregulation of two novel miRNAs (miR-191-5p and miR-197) associated with the preeclamptic condition [117]. Ayoub et al. found an upregulation for both miR-186 and miR-181a in the serum of women with pregnancy complicated by preeclampsia [136]. The dysregulated expression of circulating miRNAs depends on an epigenetic mechanism, as observed by Sekar et al. They found an increased level of miR-510 in the plasma of women with preeclamptic pregnancy correlated to a decreased methylation status of its promoter [143]. Also, C19MC has a correlation with the preeclamptic condition. Using next-generation sequencing technology, Yang et al. observed the presence of 22 different circulating dysregulated miRNAs (15 miRNAs were up-expressed and 7 miRNAs were down-expressed) in the serum of women with diagnosed preeclampsia. Among upregulated miRNAs, Yang et al. also observed three miRNAs belonging to C19MC that means miR-517, miR-518 and miR-519 [125]. The C19MC miRNAs also have been associated with the severity of the preeclamptic condition: the upregulation of miR-516-5p, miR-517, miR-520a, miR-525 and miR-526a was correlated to clinical signs of preeclampsia, requirements for the delivery and Doppler ultrasound parameters [146]. Activity research focused not only on the identification of deregulated miRNAs involved in preeclamptic pregnancy but also on the metabolic pathways that they take part in. Wu et al. not only observed dysregulated miRNAs (miR-24, miR-26a, miR-103, miR-130b, miR-181a, miR-342-3p and miR-574-5p) in the plasma of women affected by preeclampsia but revealed that they were also associated with specific signaling pathways such as the regulation of transcription and the cell cycle [122]. Khaliq’s study demonstrated that the expression level of miR-29a and miR-181a was increased in the serum of women with pregnancy complicated by preeclampsia and correlated to AKT/PI3K in the insulin signaling pathway [124]. A significant upregulation of both miR-106b and miR-326 was associated with an imbalance in T-helper type 17 (Th17)/regulatory T (Treg) cells in women affected by preeclampsia [127]. Circulating miRNAs in pregnancy can modulate several pathways, VEGF included. A study realized by Witvrouwen et al. investigated the expression level of circulating VEGF-related miRNAs in the blood samples recovered from women with diagnosed preeclampsia. The authors demonstrated that the downregulation of miR-16 in patients with preeclampsia can impair the endothelial function. Moreover, the overexpression of circulating miR-200c was increased in preeclamptic subjects and correlated with higher arterial stiffness [114]. A recent study by Nunode et al. observed a significant upregulation in the maternal serum from preeclamptic patients for miR-515-5p. In addition, bioinformatics prediction suggested a potential role of miR-515-5p in placental development: miR-515-5p could suppress trophoblast cell invasion, inhibiting the X-linked inhibitor of apoptosis protein (XIAP) which promotes cell migration by enhancing epithelial–mesenchymal transition (EMT) [107]. Kim et al. carried out an interesting study, which led to the identification of several dysregulated miRNAs associated with some clinical hallmarks of preeclampsia. In the serum of women with diagnosed preeclampsia, the expression level of miR-31-5p, miR-155-5p and miR-214-3p was significantly increased, whereas miR-1290-3p was significantly downregulated. The authors suggested that a dysregulation in the expression level of the above-mentioned miRNAs was associated with the clinical features of preeclampsia (such as hypoxia, inflammation and decreased estrogen levels) [126]. The results obtained by Kim et al. were confirmed in a study carried out by Wang et al. The authors suggested that an increased expression of miR-155 is associated with clinical manifestations of preeclampsia [135]. Also, miRNAs delivered by circulating exosomes resulted as promising biomarkers in the detection of the preeclamptic condition. Motawi et al. observed an upregulation for miRNA-136, miRNA-494 and miRNA-495 isolated from exosomes in the blood samples of women with a diagnosed preeclamptic condition [130]. Wang et al. studied the role of pregnancy-associated exosomes and their miRNA cargo miR-15a-5p in preeclamptic pathology. They isolated exosomes from the peripheral whole blood of women with pregnancy affected by preeclampsia and verified an upregulation for miR-15a-5p with respect to women with uncomplicated pregnancy [112]. Ntsethe et al. characterized the presence of exosomes in the serum of women with pregnancy complicated by preeclampsia. Exosome levels were higher in pathological pregnancies than uncomplicated ones. Both miR-155 and miR-222 moved by exosomes in the peripheral circulation were dysregulated. In fact, miR-155 was upregulated and miR-222 downregulated in the serum of women with complicated pregnancy with respect to controls [134]. Also, Aharon et al. investigated miRNAs present in exosomes. Particularly, they analyzed the blood samples of women with diagnosed preeclamptic pregnancy, isolating before exosomes and then characterizing the miRNA content. The results demonstrated a significant downregulation for miR-16-5p and miR-210 in subjects with pregnancy affected by preeclampsia with respect to healthy pregnant controls [116]. Also, the Ago-bound miR-210 was increased in the peripheral circulation of women with pregnancy complicated by preeclampsia [142].

Conversely, Luque et al. investigated miRNA circulating in maternal plasma. Evidence demonstrated that the investigated miRNAs (miR-192, miR-125b, miR-143, miR-126, miR-221, miR-942 and miR-127) were not a useful tool to predict preeclampsia, considering that their serum levels demonstrated no significant differences between the preeclampsia and control groups [153]. In the same way, Let7a-5p is not associated with preeclampsia because its expression in maternal plasma is not significantly different between uncomplicated and preeclamptic pregnancy [154]. Gunel et al. observed that the circulating level of miR-195 is not significantly different between preeclamptic and normotensive plasma samples, although miR-195 was downregulated in preeclamptic placenta samples [155].

### 5.2. miRNAs and Onset of Preeclampsia

miRNAs have been investigated to distinguish between the early (before 34 weeks of pregnancy) and late (after 34 weeks of pregnancy) onset of preeclampsia. 

Miura et al. studied the expression level of all ten miRNAs belonging to C19MC, isolated from maternal blood samples at 27–34 weeks of gestation. The authors demonstrated that, except for miR-518b and miR-519d, the remaining miRNAs were upregulated in women with an early onset of preeclampsia compared with women with a late onset of preeclampsia [145]. Dong et al. confirmed the usefulness of miRNAs as a biomarker in the detection of preeclampsia onset. Particularly, they measured the expression level of both miR-21 and miR-31 in the maternal plasma of preeclamptic women. The results demonstrated that an increase in circulating miR-31 was associated with early onset preeclampsia; meanwhile, an increase in circulating miR-21 was related to late-onset preeclampsia [119]. Decreased levels of circulating miR-126 were associated with early onset preeclampsia compared to gestation-matched controls [128]. Kolkova et al. verified the expression of miRNAs isolated from the venous blood of women with pregnancy complicated by preeclampsia. Subjects affected by preeclampsia were divided into subgroups considering the preeclampsia onset (early or late). A significant difference in circulating miRNA was observed for miR-21-5p and miR-155-5p in preeclamptic pregnancies, compared to women with uncomplicated pregnancy. The overexpression of these two miRNAs was observed in subjects with a late onset of preeclampsia compared to healthy pregnancies [120]. Whigham et al. performed a case–control study using blood samples from pregnant women. They measured the miRNA expression levels at 28 and 36 weeks of gestation from subjects who developed preeclampsia after 36 weeks’ gestation, comparing the results with levels in gestation-matched blood samples from a cohort of randomly selected controls. Notably, the expression level of miR-18a, miR-363, miR-1283, miR-149, miR-16 and miR-424 was significantly reduced in subjects who developed preeclampsia at 36 weeks’ gestation [115]. Akehurst et al. measured the miR-206 in the maternal plasma of women at 28 weeks of gestation and observed that miR-206 was upregulated in the maternal plasma of women who later developed preeclampsia [139]. Also, the miR-518b belonging to C19MC is a good biomarker for predicting pregnancy complicated by preeclampsia. Jelena et al. demonstrated that women at 24–38 weeks of gestation with an upregulation of miR-518b in plasma later developed preeclamptic pregnancy [115].

Research activity has led to the identification of miRNAs being added to the list of potential predictors for the severity of preeclampsia. Pan et al. analyzed the expression level of miRNAs in the plasma of women with normal pregnancy and pregnancy complicated by mild preeclampsia. Plasma was collected before and after parturition. They demonstrated that the parturition influenced the expression of miRNAs in the plasma of the same women and that the expression level of miR-141 and miR-221 was different between normal and preeclamptic plasma, both before and after parturition [131]. Li et al. demonstrated that the differential expression of circulating miRNAs investigated was related to the severity of the preeclamptic condition. In fact, both miR-141 and miR-29a are significantly overexpressed in the plasma of women affected by mild preeclampsia. On the contrary, miR-144 was significantly downregulated in the plasma of women affected by mild preeclampsia and severe preeclampsia with respect to uncomplicated pregnancy [123]. Birò et al. highlighted an upregulation for miR-210 in women with pregnancy complicated by severe and mild preeclampsia upon the measured level of expression in maternal plasma. However, the miR-210 expression level was not significantly different between the mild and severe preeclampsia groups [141]. Jairajpuri et al. suggested that miR-215, miR-155, miR-650, miR-210, miR-21, miR-518b and miR-29a were upregulated, and miR-18a, miR-19b1, miR-144 and miR-15b were downregulated in pregnancy complicated by severe preeclampsia versus mild preeclampsia [113].

### 5.3. miRNAs and Prediction of Preeclampsia

The ability to predict preeclampsia is a major challenge in contemporary obstetrics, and resources are now focused on the first trimester of pregnancy, where prophylactic strategies can help reduce the incidence of this disorder [156] (Figure 2).

Combined tests, such as the measurement of mean arterial pressure (MAP), the ratio of soluble Fms-like tyrosine kinase-1 to placental growth factor (sFlt-1/PlGF) and the uterine artery pulsatility index (UTPI), are already widely validated [157]. The strategy of combining biochemical and biophysical data stems from the consideration that it is unlikely that preeclampsia can be detected early by a single predictive parameter with sufficient accuracy to be clinically useful. The association between miRNAs and biophysical parameters was evaluated by demonstrating a negative correlation between miR-942 levels and maternal blood pressure and between miR-143 levels and the uterine artery Doppler pulsatility index [153]. 

A study by Luque et al. [153] indicated that the assessment of maternal serum microRNAs at the end of the first trimester of pregnancy does not appear to have any predictive value for early preeclampsia. Other studies, however, have strongly supported the need for more detailed exploration of microRNAs in the maternal circulation since they represent potential biomarkers for pregnancy-related complications [146,152,158,159]. There is a need to consider that increased placental dysfunction may stimulate the gradual release of placental mediators, including miRNAs, into the maternal circulation, leading to further diffuse maternal vascular damage and increased differential profiles of circulating miRNAs only at later stages of gestation. Studies by Winger et al. have shown that, in the first trimester of pregnancy, miRNA profiling in maternal peripheral blood mononuclear cells can successfully predict adverse outcomes, such as preeclampsia and miscarriage [129]. Circulating levels of miR-942 were lower at mid-pregnancy (12–20 weeks’ gestation) in women with preeclampsia than in the control group [151]. miR-942 might play a role in preeclampsia through ENG, an endothelial growth factor with anti-angiogenic properties [160].

Ura et al. suggested that the overexpression of miR-1233 measured at the early stages of gestation (12–14 weeks) might be a potential marker to distinguish women who later developed severe preeclampsia in the third trimester of pregnancy [152]. A subsequent study investigated the expression level of global microRNAs in first-trimester (12–14 weeks) plasma obtained from women who subsequently developed preeclampsia (at or after 34 weeks of gestation) compared to uncomplicated pregnancies. The results demonstrated that both miR-23b-5p and miR-99b-5p were downregulated in subjects with pregnancy complicated by later preeclampsia compared to the controls [121]. The panel of miRNAs that are useful in the diagnosis of preeclampsia is enriched by a study that was carried out in 2020. Li et al. investigated biological samples from women at the 12–13th week of gestation and subsequently tested the results in vitro. Analysis of the plasma samples recovered from women who later developed preeclampsia revealed a downregulation for 16 miRNAs with respect to women with uncomplicated pregnancy in the same period of gestation. Notably, the miR-125b was associated in vitro with a strong inhibition of trophoblast invasion and with the compromised activity of endothelial cells [111]. Xu et al. measured the expression level of several miRNAs in the maternal plasma of women at two different gestational weeks (15–18 weeks and 35–38 weeks). The expression level of miR-18a, miR-19b1 and miR-92a1 was reduced, while miR-210 was upregulated in the plasma of patients affected by preeclampsia compared to those in normal controls at both gestational stages [118]. In a retrospective nested cohort case–control study, 34 subjects (16 women who later developed preeclampsia and 18 women with uncomplicated pregnancy) were enrolled. Patients were invited to donate serum samples at 12, 16 and 20 weeks of gestation and at the time of preeclampsia diagnosis. The authors demonstrated that miR-628-3p was the earliest miRNA to be deregulated both at 12 and at 20 weeks of gestation and so it can be considered as a strong biomarker in preeclampsia diagnosis. Also, miR-151a-3p, miR-573 and miR-628-3p were upregulated at 16 and 20 weeks of gestation and, in the authors’ judgement, should be added to the panel of potential biomarkers for predicting pregnancy complicated by preeclampsia [132]. miR-125b was found to be significantly upregulated in the plasma of women who later developed PE, when the logical value of plasma miR-125b expression levels at the beginning of pregnancy combined with maternal age and BMI in predicting preeclampsia was 0.85 [44].

Also, miRNAs belonging to C19MC have been investigated to verify their potential role as predictors for the preeclampsia onset. A nested case–control study of a longitudinal cohort enrolled women at 10 to 13 gestational weeks. The analysis of maternal plasma demonstrated a higher level of expression for miR-517-5p in women who later developed preeclamptic pregnancy [148]. Different results emerged using microRNAs from exosomes in the plasma of women at 10–13 weeks of gestation. In fact, measurements of the expression level for miRNAs belonging to C19MC demonstrated that the downregulation of miR-517-5p, miR-520a-5p and miR-525-5p was associated with the occurrence of preeclampsia [149]. Kondracka et al. examined blood samples from women in the first trimester of gestation for the expression level of miRNAs belonging to C19MC. They reported an upregulation for miR-517 but also for miR-526 in pregnant women who later developed preeclampsia but made no significant suggestions about the possible use of miR-517-5p as a predictor of preeclampsia [147]. Analyzing the maternal serum at 12, 16 and 20 weeks of gestation, Martinez-Fierro et al. observed that increased circulating levels of 512-3p, 518f3p, 520c-3p and 520d-3p were associated with a later occurrence of preeclampsia [144]. A meta-analysis of a total of 20 studies from 8 articles including 273 patients with preeclampsia and 343 healthy pregnancies showed that circulating miRNAs could be a useful screening tool to diagnose and predict preeclampsia, with a sensitivity of 0.88 (95% CI: 0.80–0.93), a specificity of 0.87 (95% CI: 0.78–0.92) and a diagnostic odds ratio of 50.24 (95% CI: 21.28–118.62) [161]. Another meta-analysis of 14 articles, which included cyclic RNAs and miRNAs, reported a pooled AUC value of 0.86 (pooled sensitivity = 71%; pooled specificity = 84%) and a diagnostic odds ratio of 13 (95% CI: 11–19) [162].

## 6. miRNAs, Preeclampsia and Epigenetics

Preeclampsia, in addition to being one of the most frequent causes of maternal and fetal morbidity and mortality in pregnancy, has long-term negative implications for both mother and offspring [163]. Epidemiological studies indicate that women who experience preeclampsia during pregnancy have an increased vascular and metabolic risk later, as do the children of preeclamptic mothers [164]. These epidemiological considerations underlie epigenetic studies of preeclamptic disease. Although preeclampsia is a very complex disease, a great deal of evidence confirms that endothelial dysfunction is a central feature of pathogenesis and a factor that epigenetically may lead to an increased cardiovascular risk in later life. Epigenetics, or how the environment influences gene expression without altering the DNA sequence, is one of the mechanisms by which gestational hypoxia enables adaptive responses to change in the placental environment in preeclampsia. Epigenetic modifications are one of the potential mechanisms, including aberrant miRNA expression, through which the exposure to an altered environment in utero results in the development of chronic disease. The actions of miRNAs, DNA methylation and histone modification are the three most studied epigenetic processes [165]. In vitro studies have shown that miRNA expression is modulated by hypoxia, cell signaling pathways and epigenetic modifications through promoter methylation [166]. The downregulation of miRNAs also results from the hypermethylation of promoter regions [167,168]. Hypomethylation of the miR-141-3p promoter has been reported to increase the expression of miR-141-3p, which, in turn, induces inflammasome formation, a decreased expression of MMP2/9 and the inhibition of trophoblast proliferation and invasion [169]. From this point of view, the degree of miRNA methylation might have an epigenetic effect. From the perspective of endothelial dysfunction with possible epigenetic spillover, maternal and cord-derived endothelial progenitor cells (EPCs) from preeclamptic pregnancies show an aberrant miRNAs profile compared with healthy pregnancies [170]. EPCs are essential for maintaining a healthy endothelium throughout an individual’s lifetime. Decreased cell numbers and colony-forming units of maternal EPCs are described as a sign of impaired endothelial repair capacity in preeclampsia [171]. The importance of studying miRNA changes in the epigenetic domain stems from the possible therapeutic developments in preeclampsia and women’s future cardiovascular risk. The relevance of miRNAs in vascular neovascularization has been demonstrated by several knockdown approaches of enzymes involved in miRNA biogenesis [172]. Because miRNAs are known to be critical in the fine regulation and maintenance of the physiological balance of the vascular endothelium, they are targets of miRNA-based therapies through reprogramming endothelial cells.

## 7. Conclusions

Several non-coding RNAs and, thus, miRNAs are differentially expressed in the pathophysiology of women’s health in general [173] and in pregnancy and the placenta in particular. In recent years, the number of identified miRNAs has significantly increased; however, the exact mechanisms remain to be elucidated, mainly because of the cell-specific functions exhibited by many miRNAs. A deeper understanding of miRNAs and their relationships with gene modifications will help in determining the mechanism by which these molecules contribute to placental development. The identification of miRNAs that may act as potential non-invasive biomarkers for the prediction of pregnancy outcomes in the first trimester, especially among high-risk women, may have implications for research, identifying signaling pathways for further investigation and clinical implications, facilitating early diagnosis and timely interventions. This challenge is not easy to address since preeclampsia is probably not a single disease but may present in several different forms, and there are many difficulties in finding, measuring and reproducing miRNA results. From this review, we see the possibility that, in the near future, molecular biology may, through the application of gene therapy on miRNAs, intervene in the pathological basis of preeclampsia from the very early stages of placental implantation and development.

## Figures and Tables

**Figure 1 ijms-25-01418-f001:**
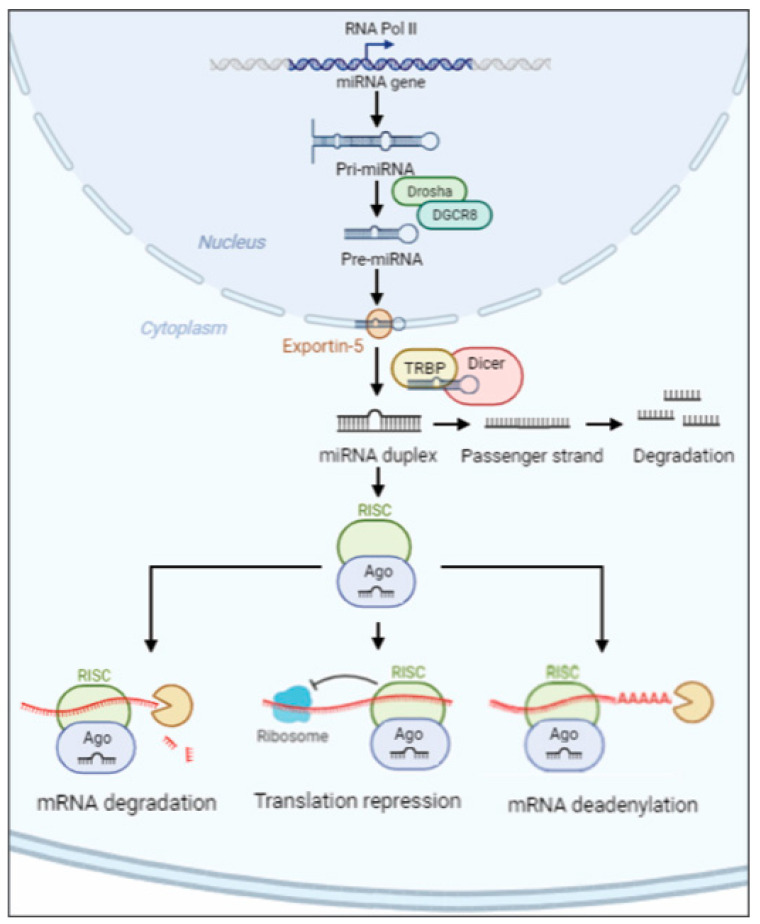
Schematic overview of miRNA biosynthesis. The miRNA pathway produces pri-miRNA transcripts from miRNA genes encoded in exonic, intronic or intergenic regions. In the nucleus, Drosha and DGCR8 digest the pri-miRNA into pre-miRNA. The pre-miRNA is driven into the cytoplasm by the complex Exportin-5–Ran-GTP. Once the mature length is obtained, the functional strand is loaded with Ago2 protein into RISC complex which silences the mRNA’s target through mRNA deadenylation, translational repression or degradation.

**Figure 2 ijms-25-01418-f002:**
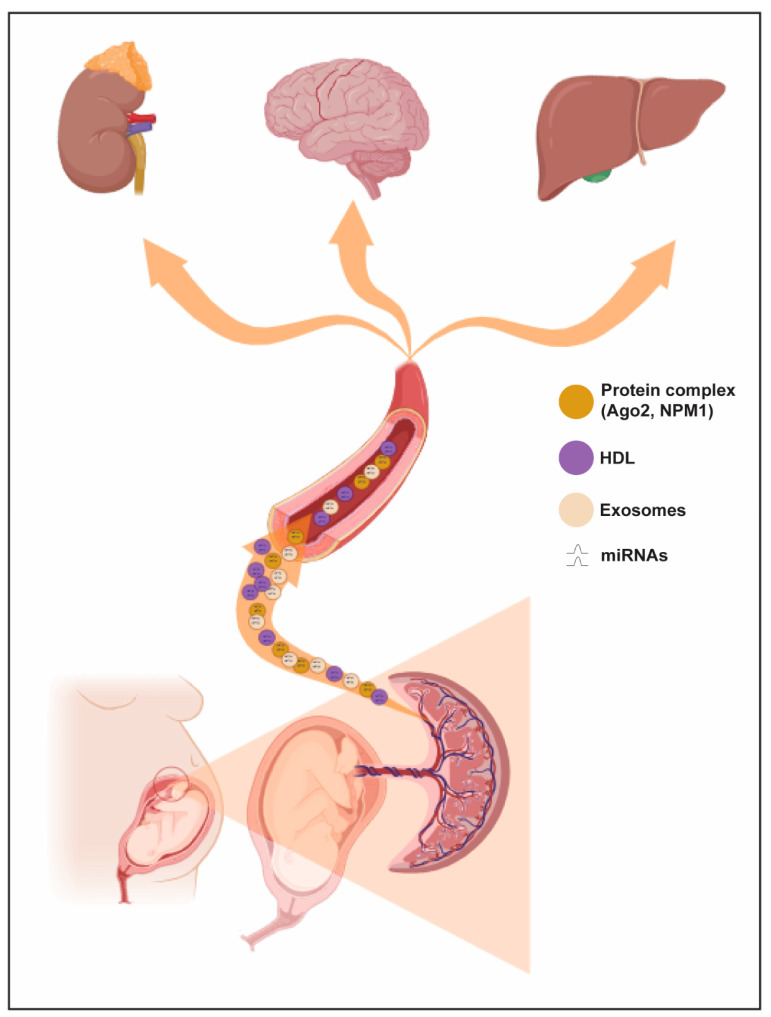
Trafficking routes of circulating miRNAs during pregnancy. Circulating miRNAs may be delivered to maternal circulation and affect various cell events in maternal targeting organs (e.g., kidney, brain and liver).

**Table 1 ijms-25-01418-t001:** miRNAs significantly dysregulated in clinical diagnosis of preeclampsia.

miRNAs	Diagnosed PE	Early Onset PE	Late Onset PE	Mild PE	Severe PE	Prediction of PE	Ref.
**miR-15a-5p**							[111]
						[112]
**miR-15b**							[113]
**miR-16**							[114]
						[115]
**miR-16-5p**							[116]
**miR-17-5p**							[117]
**miR-18a**							[118]
						[113]
						[115]
**miR-19b1**							[118]
						[113]
**miR-21**							[119]
						[113]
**miR-21-5p**							[120]
**miR-22-5p**							[111]
**miR-24-3p**							
**miR-23b-5p**							[121]
**miR-24**							[122]
**miR-26a**							[122]
**miR-29a**							[123]
						[124]
						[113]
						[125]
**miR-29a-3p**							[117]
**miR-31**							[119]
**miR-31-5p**							[126]
**miR-92a-1**							[118]
**miR-92-a-1-3p**							[118]
**miR-93-5p**							[111]
**miR-99-5p**							[121]
**miR-103**							[122]
**miR-106a**							[111]
**miR-106b**							[127]
**miR-125b**							[111]
						[44]
						[125]
**miR-125a-5p**							[125]
**miR-126**							[128]
**miR-126-3p**							[111]
**miR-130a-3p**							[111]
**miR-130b**							[122]
**miR-132**							[129]
**miR-132-3p**							[117]
**miR-133b**							[129]
**miR-136**							[130]
						[125]
**miR-141**							[131]
						[123]
**miR-144**							[123]
						[113]
**miR-146a**							[129]
**miR-149**							[115]
**miR-151a-3p**							[132]
**miR-152**							[133]
**miR-155**							[113]
						[134]
						[135]
**miR-155-5p**							[126]
						[120]
**miR-181a**							[122]
						[122]
						[136]
**miR-185**							[125]
**miR-186**							[136]
**miR-191-5p**							[111]
						[117]
**miR-195-5p**							[137]
**miR-196b**							[138]
**miR197-3p**							[117]
**miR-200c**							[114]
**miR-204-3p**							[111]
**miR-206**							[139]
						[140]
**miR-210**							[133]
						[117]
						[118]
						[141]
						[113]
						[142]
						[116]
						[129]
**miR-214-3p**							[126]
**miR-215**							[113]
**miR-218-5p**							[117]
**miR-221**							[131]
**miR-218-5p**							[117]
**miR-223**							[125]
**miR-302b-3p**							[117]
**miR-320c**							[125]
**miR-326**							[127]
**miR-328**							[117]
**miR-342-3p**							[122]
**miR-363**							[115]
**miR-365a-3p**							[111]
**miR-374a-5p**							[111]
**miR-375**							[117]
**miR-424**							[115]
**miR-494**							[130]
**miR-495**							[130]
**miR-510**							[143]
**miR-512-3p**							[144]
**miR-515-5p**							[145]
						[107]
**miR-516-5p**							[146]
**miR-516a-5p**							[146]
**miR-516b**							[145]
**miR-517**							[146]
						[147]
**miR-517-5p**							[148]
						[149]
**mirR-517b**							[125]
**miR-517c**							[125]
**miR-518b**							[113]
						[115]
**miR-518e**							[125]
**miR-518f3p**							[144]
**miR-519a**							[125]
**miR-519d**							[125]
**miR-520-5p**							[145]
**miR-520a**							[146]
**miR-520a-5p**							[149]
**miR-520c-3p**							[144]
**miR-520d-3p**							[144]
**miR-520g**							[125]
**miR-520h**							[145]
						[125]
**miR-521**							[125]
**miR-525**							[146]
**miR-525-5p**							[145]
						[149]
**miR-526**							[145]
						[147]
**miR-559-5p**							[111]
**miR-526a**							[146]
**miR-542-3p**							[125]
**miR-573**							[132]
**miR-574-5p**							[122]
						[111]
**miR-628-3p**							[132]
**miR-650**							[113]
**miR-885-5p**							[150]
**miR-942**							[151]
**miR-1229p**							[129]
**miR-1233**							[152]
**miR-1244**							[129]
**miR-1260**							[125]
**miR-1272**							[125]
**miR-1283**							[115]
**miR-1290-3p**							[126]
**miR-1323**							[145]
**miR-4264-5p**							[111]
**Let-7a-5p**							[111]
**Let-7a**							[125]
**Let-7d**							[125]
**Let-7f**							[125]
**Let-7f-1**							[125]

Abbreviations: PE: preeclampsia. 

 Up-regulated; 

 Down-regulated.

## Data Availability

No new data were created or analyzed in this study. Data sharing is not applicable to this article.

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
