# Peer review of "Circulating miRNAs and Preeclampsia: From Implantation to Epigeneticsâ€"

_ijms, 2024, doi:10.3390/ijms25031418_

Round 1
Reviewer 1 Report
Comments and Suggestions for Authors
Summary: This paper reviews the literature on extracellular vesicles as transporters of biologically important molecules during pregnancy with particular attention to preeclampsia. The paper makes an incremental contribution to our understanding of ECV in pregnancy, since many similar reviews have been published in recent years. There is nothing inherently wrong with the paper, and it could be seen as an updated assessment of the field. There are a few suggestions that could strengthen the contribution.
1. The sections could be collapsed and consolidated, since there is considerable overlap.
2. There are many instances where hyphenated words are found that do not require such, and may likely result from editing issues.
3. It would be helpful to improve the overall English of the paper.
4. The outline of table is very confusing and could easily be improved.
Comments on the Quality of English LanguageNeed attention to English writing and presentation.
Author Response
COMMENT
The sections could be collapsed and consolidated, since there is considerable overlap.
RESPONSE
Paragraph structure has been rearranged to avoid overlap and repetition.
COMMENT
There are many instances where hyphenated words are found that do not require such, and may likely result from editing issues.
RESPONSE
The text has been fully revised for spelling errors.
COMMENT
It would be helpful to improve the overall English of the paper.
RESPONSE
The text was submitted to and revised by MDPI's English editing service
COMMENT
The outline of table is very confusing and could easily be improved.
RESPONSE
As suggested Table 1 has been improved in appearance and understanding.
Reviewer 2 Report
Comments and Suggestions for Authors
The paper submitted for review deals with a very important and at the same time challenging topic, which influences on many aspects of pregnancy and overall women's health, which is preeclampsia. The manuscript is a review analyzing the role of miRNA in the development of the preeclampsia and the possibility of using miRNA as a diagnostic tool and a potential therapeutic target. The manuscript comprehensively presents the latest scientific data. I only have a few minor comments. I am convinced that after taking these comments into account, the article can be published in IJMS.
1) Please include Materials and Methods section - this is scoping review, systematic review? Which are the exclusion/inclusion factors in article selection?
2) Please cite relevant reference which impose miRNA role in other gynecological conditions, such as PCOS (refer to PMID: 36009364)
3) A schematic figure showing the functional role of microRNAs in a cell would be a nice addition to the reading experience.
Comments on the Quality of English LanguageModerate editing of English language required.
Author Response
COMMENT
Please include Materials and Methods section - this is scoping review, systematic review? Which are the exclusion/inclusion factors in article selection?
RESPONSE
A "Materials and Methods" section has been included in the text to indicate the type of publication and the methods used to select the studies.
COMMENT
Please cite relevant reference which impose miRNA role in other gynecological conditions, such as PCOS (refer to PMID: 36009364)
RESPONSE
The suggested reference was included to emphasize the importance of miRNAs in all female gynecological pathology.
COMMENT
A schematic figure showing the functional role of microRNAs in a cell would be a nice addition to the reading experience.
RESPONSE
As suggested, Figure 1 on the cellular mechanisms of circulating miRNAs has been included.
COMMENT
Moderate editing of English language required.
RESPONSE
The text was submitted to and revised by MDPI's English editing service
Round 2
Reviewer 1 Report
Comments and Suggestions for Authors
Acceptable
Comments on the Quality of English LanguageThe English language needs some work.